# Electron Density and Effective Atomic Number of Normal-Appearing Adult Brain Tissues: Age-Related Changes and Correlation with Myelin Content

**DOI:** 10.3390/tomography11090095

**Published:** 2025-08-25

**Authors:** Tomohito Hasegawa, Masanori Nakajo, Misaki Gohara, Kiyohisa Kamimura, Tsubasa Nakano, Junki Kamizono, Koji Takumi, Fumitaka Ejima, Gregor Pahn, Eran Langzam, Ryota Nakanosono, Ryoji Yamagishi, Fumiko Kanzaki, Takashi Yoshiura

**Affiliations:** 1Department of Radiology, Kagoshima University Graduate School of Medical and Dental Sciences, 8-35-1 Sakuragaoka, Kagoshima 890-8544, Japan; k0314697@kadai.jp (T.H.); k0707094@kadai.jp (M.N.); akuta_dear2@icloud.com (M.G.); kiyohisa@m2.kufm.kagoshima-u.ac.jp (K.K.); tnakano@m.kufm.kagoshima-u.ac.jp (T.N.); operation.desert-fox.xfa-27@hotmail.co.jp (J.K.); takumi@m2.kufm.kagoshima-u.ac.jp (K.T.); a3333333@m.kufm.kagoshima-u.ac.jp (F.E.); k0120951@kadai.jp (R.N.); k3770720@kadai.jp (R.Y.); k2628108@kadai.jp (F.K.); 2Philips GmbH Market DACH, Röntgenstr. 22, 22335 Hamburg, Germany; gregor.pahn@philips.com; 3Philips Healthcare, Nahum Het Street 16, Haifa 3100202, Israel; eran.langzam@philips.com

**Keywords:** dual-energy computed tomography (DECT), electron density, effective atomic number (Z_eff_), myelin, age-related change

## Abstract

**Objectives:** Few studies have reported in vivo measurements of electron density (ED) and effective atomic number (Z_eff_) in normal brain tissue. To address this gap, dual-energy computed tomography (DECT)-derived ED and Z_eff_ maps were used to characterize normal-appearing adult brain tissues, evaluate age-related changes, and investigate correlations with myelin partial volume (V_my_) from synthetic magnetic resonance imaging (MRI). **Materials and Methods:** Thirty patients were retrospectively analyzed. The conventional computed tomography (CT) value (CT_conv_), ED, Z_eff_, and V_my_ were measured in the normal-appearing gray matter (GM) and white matter (WM) regions of interest. V_my_ and DECT-derived parameters were compared between WM and GM. Correlations between V_my_ and DECT parameters and between age and DECT parameters were analyzed. **Results:** V_my_ was significantly greater in WM than in GM, whereas CT_conv_, ED, and Z_eff_ were significantly lower in WM than in GM (all *p* < 0.001). Z_eff_ exhibited a stronger negative correlation with V_my_ (ρ = −0.756) than CT_conv_ (ρ = −0.705) or ED (ρ = −0.491). ED exhibited weak to moderate negative correlations with age in nine of the 14 regions. In contrast, Z_eff_ exhibited weak to moderate positive correlations with age in nine of the 14 regions. CT_conv_ exhibited negligible to insignificant correlations with age: **Conclusions:** This study revealed distinct GM–WM differences in ED and Z_eff_ along with opposing age-related changes in these quantities. Therefore, myelin may have substantially contributed to the lower Z_eff_ observed in WM, which underlies the GM–WM contrast observed on non-contrast-enhanced CT.

## 1. Introduction

In normal adult brains, the gray matter (GM) exhibits higher computed tomography (CT) attenuation (measured in Hounsfield units) than the white matter (WM). Loss of the gray–white contrast is a key non-contrast-enhanced CT finding of hyperacute ischemic stroke [1,2]. Brooks et al. [3] attributed this native gray–white contrast on CT to the differences in the chemical composition of the two brain compartments. With higher lipid content and lower water content, the WM includes more carbon, which is lower in atomic number (i.e., 6), and less oxygen, which is higher in atomic number (i.e., 8), than the GM. Hence, photoelectric X-ray absorption is smaller in the WM, resulting in lower CT value in the WM than in the GM. Recently, dual-energy CT (DECT) has become widely available in clinical practice. DECT enables the imaging of electron density (ED) and effective atomic number (Z_eff_), which represents a weighted average of the atomic numbers of a material’s constituent elements [4,5,6,7,8,9,10]. However, publications on the ED and Z_eff_ in the normal human brain obtained using clinical CT scanners are limited. To the best of our knowledge, age-related changes have not been reported. With the clinical implementation of photon-counting detector CT underway, normal brain ED and Zeff values, as well as their age-related changes, may serve as valuable fundamental data for future investigations into the potential of CT-derived parameters for the quantitative evaluation of cerebral diseases. Myelin is a key contributor to the compositional differences between the GM and WM. Myelin is a lipid-rich material that covers neuronal axons and serves as an electric insulator [11,12]. It enables fast and efficient signal transmission via saltatory nerve conduction [11]. Myelin has a strong impact on magnetic resonance imaging (MRI) contrast. For example, myelin is associated with the characteristic properties of longitudinal and transverse relativity and magnetization transfer [13,14,15]. Taking advantage of the characteristic MR properties of myelin, a multiparametric model for myelin content based on T1, T2, and proton density (PD) values derived from synthetic MRI data were proposed [16]. The quantitative evaluation of myelin partial volume (V_my_) based on this model was validated through a correlation with measurement in cadaveric brain tissue [17] and has been used to assess various brain diseases [18,19,20,21,22]. To date, no studies have examined the correlation between Z_eff_ and ED derived from DECT and V_my_ derived from synthetic MRI. We hypothesized that myelin exhibits a major impact on the atomic number difference, which results in gray–white contrast on non-contrast-enhanced brain CT; thus, Z_eff_ is strongly correlated with V_my_. The purposes of this study were to characterize ED and Z_eff_ in normal-appearing adult brain tissues and their age-related changes in vivo using a clinical DECT scanner and to clarify the correlations between DECT-derived ED or Z_eff_ and V_my_ derived from synthetic MRI.

## 2. Materials and Methods

### 2.1. Study Population

This retrospective study was approved by the Institutional Review Board of our institution on 30 July 2024 (approval number 230267). The requirement for informed consent was waived considering the retrospective and noninvasive nature of the study. All procedures were conducted according to the principles outlined in the Declaration of Helsinki.

In total, 112 consecutive patients who underwent DECT and synthetic MRI as part of the evaluation of intracranial diseases between April 2021 and March 2023 at our hospital were considered for inclusion. The inclusion criteria were as follows: (1) age older than 18 years and (2) interval between DECT and synthetic MRI within 30 days. The exclusion criteria were as follows: (1) present or past history of neurological or psychiatric diseases, (2) history of neurosurgical intervention, chemotherapy, or brain radiation therapy; (3) poor quality of CT and MR images; (4) presence of supratentorial intra-axial lesions, such as infarctions, contusion, and tumors, on MRI; (5) severity of cerebral hyperintensity on fluid-attenuated inversion recovery (FLAIR) images of grade 3 or higher either or both in the periventricular or deep WM according to the classification by Fazekas et al. [23]; (6) presence of ≥4; microbleeds on susceptibility-weighted images; (7) presence of midline shift or extensive brain edema; and (8) presence of ventricular enlargement suspected of hydrocephalus. A board-certified radiologist with 30 years of experience in neuroradiology assessed the exclusion criteria. The images and clinical information of the patients were obtained from the picture archiving and communication system and medical charts in our hospital.

### 2.2. CT Imaging

CT images were obtained using a multidetector row DECT system (IQon Spectral CT; Philips Healthcare, Best, The Netherlands), which was equipped with a dual-layer spectral detector and a single X-ray source, constantly enabling the acquisition of high- and low-energy projection data and retrospective dual-energy data analysis. The imaging parameters were as follows: tube voltage, 120 kVp; effective tube current–exposure time product, 400 mAs; detector-row configuration, 64 × 0.625 mm; gantry rotation time, 0.4 s; and pitch, 0.36. The reconstruction slice thickness was 1.0 mm. The volume CT dose index (CTDI_vol_) was 67.7 mGy; the mean dose–length product (DLP) was 1513.0 mGy·cm (range, 1349–1824 mGy·cm). Because a dual-layer detector system was used, no additional radiation exposure was required to obtain dual-energy data. The CT systems were inspected every three months by Philips service engineers, and underwent daily phantom-based quality control checks performed by the facility’s radiologic technologists.

Conventional 120-kVp CT images (CT_conv_) and quantitative ED and Z_eff_ maps were generated using a dedicated workstation (IntelliSpace Portal; Philips Healthcare) [24]. ED was expressed in percent ED relative to water (%EDW).

### 2.3. MR Imaging

MRI scans were obtained using a 3.0-T system (Achieva dStream; Philips Healthcare, Best, The Netherlands) with a 32-channel head receiving coil. Synthetic MRI data were acquired as a part of routine brain MRI examination using a two-dimensional axial method with quantification of relaxation times and PD by multi-echo acquisition of saturation-recovery using turbo spin-echo readout pulse sequence. It used two echo times (13 and 100 ms) and four delay times (110, 440, 1210, and 2530 ms) to generate eight actual and eight imaginary images. Other parameters were as follows: repetition time, 3600 ms; flip angle, 90°; sensitivity encoding factor, 2.2; field of view, 230 × 230 mm^2^; matrix resolution, 315 × 258; echo-train length, 10; slice thickness, 5.0 mm; gap, 1.0 mm; 24 slices; and scan time, 5 min and 24 s. V_my_ was mapped using SyMRI (version 19.3; SyMRI, Linköping, Sweden). This software also generated synthetic T1-weighted images, T2-weighted images, PD, and FLAIR images in addition to quantitative maps of T1, T2, PD, and V_my_ [16,25]. In addition to synthetic MRI, conventional MR images were acquired, including pre-contrast 2D FLAIR and 3D susceptibility-weighted images. Table 1 lists the imaging parameters of those images.

### 2.4. Image Analysis

For each patient, CT images (CT_conv_, ED, and Z_eff_) were registered with the corresponding synthetic T2WI using Statistical Parametric Mapping 12 (revision 7219; Functional Imaging Laboratory, UCL Queen Square Institute of Neurology, London, UK; https://www.fil.ion.ucl.ac.uk/spm/ (accessed on 22 August 2025)). Two independent radiologists with 9 and 18 years of experience in diagnostic radiology (TH and MN, respectively) measured the regions of interest (ROIs) using ImageJ (version 1.52a; National Institutes of Health, Bethesda, MD, USA; https://imagej.net/ij/ (accessed on 22 August 2025)). Ovoid ROIs (area 15 mm^2^) were manually placed on the synthetic PD images in 12 GM (bilateral caudate heads, putamina, globus pallidi, and lateral, medial, and dorsal thalami) and 14 WM (bilateral frontal, parietal, temporal, and occipital lobes, bilateral centrum semiovales [CSs] and posterior limbs of the internal capsule [PLIC], and genu [GCC] and splenium [SCC] of the corpus callosum) regions and cerebrospinal fluid (CSF) in the bilateral lateral ventricles (Figure 1, Table 2). ROIs were set carefully to exclude partial volume averaging at tissue boundaries. Areas with calcification on CT and abnormal signal intensity on FLAIR or susceptibility weighted imaging (SWI) were avoided. The ROIs were copied onto the corresponding maps of V_my_, CT_conv_, ED, and Z_eff_, in which the ROI mean parametric values were obtained.

### 2.5. Statistical Analysis

Statistical analyses were performed using Statistical Package for the Social Sciences (version 28; IBM Corp.; Armonk, NY, USA; https://www.ibm.com/products/spss-statistics, (accessed on 23 August 2025)). *p*-values <0.05 were used to denote statistical significance. Interobserver agreement for the ROI measurement was evaluated using the intraclass correlation coefficient (ICC). ICC values of 0.00–0.20, 0.21–0.40, 0.41–0.60, 0.61–0.80, and 0.81–1.00 indicate poor, fair, moderate, good, and excellent agreements, respectively. Furthermore, Bland–Altman analysis was used to report the bias and 95% limits of agreement between the two observers. For parameters with no significant interobserver differences, the averaged values from the two observers were used for further analyses. Comparisons between numerical variables were performed using the Mann–Whitney U test or unpaired *t*-test, as appropriate. The Shapiro–Wilk normality test was used to check the normality of the data. Spearman’s correlation coefficient (ρ) was used to analyze correlations between the V_my_ and CT parameters and between patient age and CT parameters and V_my_. For the latter, data from the right and left ROIs were averaged, except for those from the GCC and SCC. Because of the limited patient age range, quadratic fitting was not performed. Correlation coefficients of 0.00–0.19, 0.20–0.39, 0.40–0.69, 0.70–0.89, and 0.90–1.00 indicated negligible to very weak, weak, moderate, strong, and very strong correlations, respectively (modified from Schober et al. [26]). Simple and multiple regression analyses were performed to generate a mathematical model of V_my_ using CT-derived parameters, with the coefficient of determination (R^2^) used to assess the goodness of fit—that is, the proportion of variance in V_my_ explained by the model.

## 3. Results

### 3.1. Participants

Figure 2 presents the patient selection flowchart. Finally, 30 patients were eligible for the analysis (Table 3). The patients’ ages ranged from 35 to 84 years. Twenty-two (73%) patients were female. The most frequent indication for imaging examinations was extra-axial tumors (23 patients, 77%), of which 20 (67%) were pathologically proven to be meningiomas.

### 3.2. Interobserver Agreement of the Measurements

Table 4 presents the agreement between the two observers. The ICCs between the two readers were 0.975 for V_my_, 0.873 for CT_conv_, 0.884 for ED, and 0.948 for Z_eff_, indicating excellent agreement. The Bland–Altman plot revealed reasonable agreement for each parameter. No obvious bias was observed for any parameter. Figure 3 presents the Bland–Altman plot for each imaging parameter.

### 3.3. Comparison of V_my_ and CT Parameters Between the WM and GM

Figure 4 presents representative images of V_my_, CT_conv_, ED, and Z_eff_. Table 5 present the values of V_my_ and CT parameters in the WM, GM, and CSF. V_my_ was significantly greater in the WM (35.6 ± 4.7%) than in the GM (14.0 ± 8.1%) (*p* < 0.001), whereas CT_conv_, ED, and Z_eff_ were significantly lower in the WM (25.9 ± 2.5 HU, 102.8 ± 0.2%EDW, and 7.2 ± 0.0, respectively) than in the GM (33.4 ± 3.2 HU, 103.2 ± 0.2%EDW, and 7.3 ± 0.1, respectively) (each *p* < 0.001). CT_conv_ and ED were significantly lower in the CSF (4.1 ± 2.4 HU and 100.2 ± 0.2%EDW, respectively) than in both the WM and GM (each *p* < 0.001). Z_eff_ was significantly greater in the CSF (7.3 ± 0.1) than in the WM (*p* < 0.001) but was similar to that in the GM.

### 3.4. Correlation Between V_my_ and CT Parameters and Regression Analyses

Table 6 and Figure 5 present the correlation between V_my_ and CT parameters. CT_conv_, ED, and Z_eff_ were negatively correlated with V_my_ (each *p* < 0.001) when all WM and GM ROIs were included in the analysis. The strongest correlation was observed for Z_eff_ (ρ = −0.756), followed by CT_conv_ (ρ = −0.705). The correlation between V_my_ and ED was moderate (ρ = −0.491). The three CT parameters were also negatively correlated with V_my_ in the GM: the correlation was moderate for Z_eff_ (ρ = −0.478), weak for CT_conv_ (ρ = −0.379), and negligible to very weak for ED (ρ = −0.151). In the WM, only ED was weakly and positively correlated with V_my_ (ρ = 0.202).

Table 7 summarizes the regression analysis results. Simple regression analysis for all ROIs revealed that Z_eff_ had the highest model fit (R^2^ = 0.606) among the three CT parameters. In the multiple regression analysis, Z_eff_ and ED were included as independent variables and increased the model’s explanatory power (R^2^ = 0.675).

### 3.5. Correlation Between Patient Age and V_my_ or CT Parameters

Table 8 presents Spearman’s correlation coefficients between patient age and imaging parameters. V_my_ exhibited a significant negative correlation with age in 10 of 14 regions. A stronger correlation was noted in WM regions, including the frontal (ρ = −0.741) and parietal (ρ = −0.719) lobes and CS (ρ = −0.701), followed by moderate correlations in the occipital (ρ = −0.678) and temporal WM (ρ = −0.649), globus pallidus (ρ = −0.646), lateral thalamus (ρ = −0.605), PLIC (ρ = −0.552), GCC (ρ = −0.413), and medial thalamus (ρ = −0.378). The correlation between patient age and CT_conv_ was either negligible to very weak or weak without statistical significance in all regions. ED exhibited a weak to moderate negative correlation with age in nine of the 14 regions. Statistical significance was noted in three WM regions (i.e., CS, PLIC, and occipital lobe; ρ ranged from −0.411 to −0.450). In contrast, Z_eff_ exhibited a weak to moderate positive correlation with patient age in nine of the 14 regions. A significant moderate correlation was observed in the WM of the occipital (ρ = 0.549), temporal (ρ = 0.437), and frontal (ρ = 0.405) lobes. Figure 6 presents age-related changes in V_my_, CT_conv_, ED, and Z_eff_ in the WM regions.

## 4. Discussion

This study revealed that, in normal-appearing adult brain tissues, CT_conv_, ED, and Z_eff_ were significantly lower in the WM than in the GM. CT_conv_ was strongly and negatively correlated with V_my_ derived from synthetic MRI, and Z_eff_ derived from DECT exhibited a stronger correlation with V_my_ than CT_conv_ and ED. These results support our hypothesis that myelin exhibits a major impact on the atomic number difference, which results in the gray–white contrast on non-contrast-enhanced brain CT. ED and Z_eff_ exhibited negative and positive correlations with age, respectively, particularly in the WM regions, whereas CT_conv_ exhibited no obvious age-related changes in both the GM and WM.

Although DECT has been widely used in clinical practice, reports on ED and Z_eff_ in the cerebral GM, WM, and CSF of living human brains are limited. The mean ED was significantly lower in the WM than in the GM; however, the difference was smaller (approximately 0.4%EDW) than the difference between the WM and CSF (approximately 2.6%EDW). These findings are consistent with those of a previous study on the calculated ED values of the respective tissues [27], although their absolute values were slightly greater than ours. The mean Z_eff_ was significantly lower in the WM than in the GM, and their difference was approximately 0.1 (1.5%). This is also qualitatively consistent with the report of the calculated results [27]. The lack of a significant difference in Z_eff_ between the CSF and GM appears to contradict the values previously reported by Hünemohr et al. [27] (7.45 for the CSF and 7.34 for the cerebral GM), which were calculated with six major elements (i.e., H, C, N, O, Ca, and P). However, when trace elements (i.e., Na, S, Cl, and K) were considered, the calculated Z_eff_ values in the CSF and GM were similar [24,28], supporting our findings.

X-ray absorption includes photoelectric and Compton components. Brook et al. [3] reported that the gray–white contrast on non-contrast-enhanced CT originates from the photoelectric component, which is influenced by the atomic number of the sample; however, there is no difference in the Compton component between the two compartments. Based on a quantitative simulation using the reported values, the gray–white difference in the atomic number was attributable to the relative abundance of carbon and the relative paucity of oxygen in the WM, which has more lipid and less water content than the GM [3]. Myelin mainly comprises a complex lipid called sphingolipid. These substances possess many carbon atoms in their carbon chains. The strong negative correlation between Z_eff_ and V_my_ revealed in this study may be caused by such chemical structural features of myelin.

In our results, the mean ED was slightly lower in the WM than in the GM. Because the EDs of water and lipid are substantially lower than those of brain tissues [27], the ED of brain tissues can be affected by both the lipid and water content. The lower water content in the WM than in the GM does not explain the lower ED in the WM. Thus, it is likely due to the relative abundance of lipid content in the WM versus the GM.

Our regression analysis revealed that Z_eff_ explained a greater proportion of the variance between V_my_ and ED or CT_conv_. Thus, Z_eff_ is a potential CT imaging marker of myelin in normal-appearing adult brains. In the multiple regression analysis, a combination of Z_eff_ and ED improved the explanatory power of the model (R^2^ = 0.689) compared with Z_eff_ alone (R^2^ = 0.617). As predicted by the theory proposed by Brooks et al. [3], the Z_eff_ of brain tissues is influenced by both the lipid and water content. We speculate that the improvement in modeling of V_my_ by adding ED to Z_eff_ could be attributable to the correction of the water content. Thus, in normal-appearing adult brains, the myelin content can be estimated with some degree of accuracy by combining Z_eff_ and ED.

V_my_ exhibited negative correlations with age throughout the GM and WM, more strongly in the cerebral WM. This is in accordance with a previous synthetic MRI study by Hagiwara et al. [29] and likely represents age-related demyelination. CT_conv_ exhibited no obvious correlation with age, which is consistent with the findings in several previous studies [30,31]. In contrast, ED exhibited weak to moderate negative correlations and Z_eff_ exhibited weak to moderate positive correlations with age, mainly in the WM. The positive correlation of Z_eff_ with age can be attributable to the synergetic effects of age-related demyelination and axonal loss and the accompanying increase in interstitial water. The negative correlation of ED with age indicates increased water content. The lack of age-related changes in CT_conv_ may be attributable to the opposing age-related effects of ED and Z_eff_ on CT_conv_. Thus, ED and Z_eff_ can detect age-related WM changes that are undetectable by the conventional CT value.

DECT parameters acquired on clinical CT scanners correlated significantly with V_my_. This association suggests that DECT may serve as a noninvasive, adjunctive tool for evaluating disorders characterized by myelin loss, including neurodegenerative and demyelinating conditions traditionally assessed with MRI. Compared with MRI, CT is less costly, faster, and operationally simpler; moreover, DECT remains feasible when MRI is contraindicated or difficult (e.g., in patients with non–MR-conditional metallic implants, severe claustrophobia, or in pediatric and older patients for whom sedation is challenging), and may function as an alternative in selected scenarios. Furthermore, the result that ED and Z_eff_ correlate with age even within normal-appearing white matter on MRI suggests that these measures may enable earlier detection of age-related changes in brain parenchyma. Given their quantitative nature, establishing age-stratified reference ranges for ED and Z_eff_ could improve the sensitivity for detecting abnormalities.

This study has several limitations that should be considered. First, because of radiation exposure, healthy volunteers were not scanned. The number of subjects was small because only patients who underwent both DECT and synthetic MRI were included and patients with any supratentorial lesions on MRI that could profoundly affect the measurements of the imaging parameters were excluded. Younger patients were not included because CT imaging is typically avoided for those who are more sensitive to radiation exposure than older adults. It consequently limited the age range (≥35 years); these circumstances denied us the opportunity to observe continuing myelination in young adults. The strong female predominance in the patient group of this study is likely related to the fact that many patients had meningiomas. This bias prevented the sex-specific analysis of the imaging parameters. Finally, this study did not examine the relationship between ED or Z_eff_ and the water content in brain tissue. As mentioned earlier, in addition to lipid content, water content has been postulated as a contributor to Z_eff_ difference between the GM and WM [3]. Using a synthetic MRI-based approach, we clarified the correlation between Z_eff_ and lipid-rich myelin. However, images of water content were not available in our retrospective analysis. In future studies, the inclusion of imaging techniques sensitive to water content, for example, based on diffusion MRI [32], could help clarify the complete picture of the relationship between CT-derived parameters (Z_eff_ and ED) and brain tissue composition.

## 5. Conclusions

In conclusion, this study confirmed the differences in ED and Z_eff_ between the GM and WM using a clinical DECT scanner. Moreover, a strong negative correlation was observed between Z_eff_ and myelin content derived from synthetic MRI, suggesting that abundant carbon in myelin is a dominant contributor to lower Z_eff_ in the WM than in the GM that generates the gray–white contrast on non-contrast-enhanced CT images of adults. Unlike CT_conv_, ED and Z_eff_ exhibited opposing age-related changes mainly in the WM, which may offer a novel opportunity for CT-based evaluation of WM aging.

## Figures and Tables

**Figure 1 tomography-11-00095-f001:**
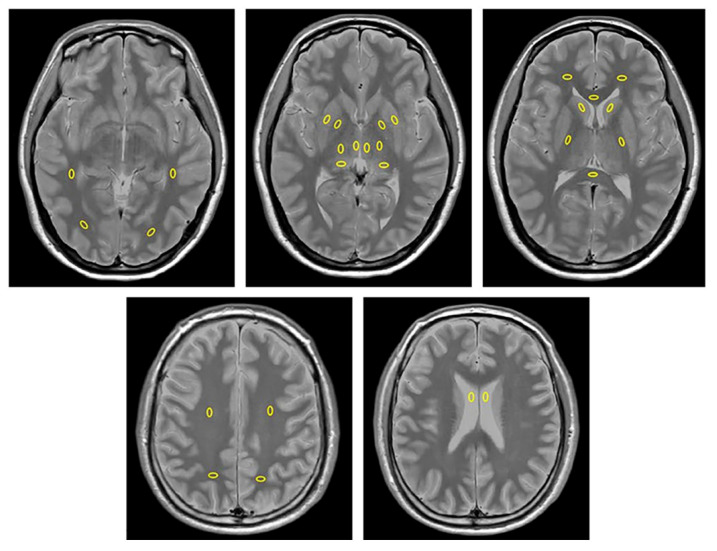
Regions of interests (ROIs) placed on synthetic proton density images. In the frontal lobe, the ROIs were placed anterior to the frontal horn of the lateral ventricle. In the temporal lobe, the ROIs were placed in the WM lateral to the inferior horn of the lateral ventricles. In the occipital lobe, the ROIs were positioned in the WM near the occipital pole. In the parietal lobe, the ROIs were placed in the WM at the level of the centrum semiovale. Within the thalamus, three ROIs were placed bilaterally in the presumed medial nuclei, posterior region (pulvinar), and lateral nuclei (excluding the pulvinar). The ROIs in the lateral ventricle were positioned anteriorly in the body to avoid choroid plexus.

**Figure 2 tomography-11-00095-f002:**
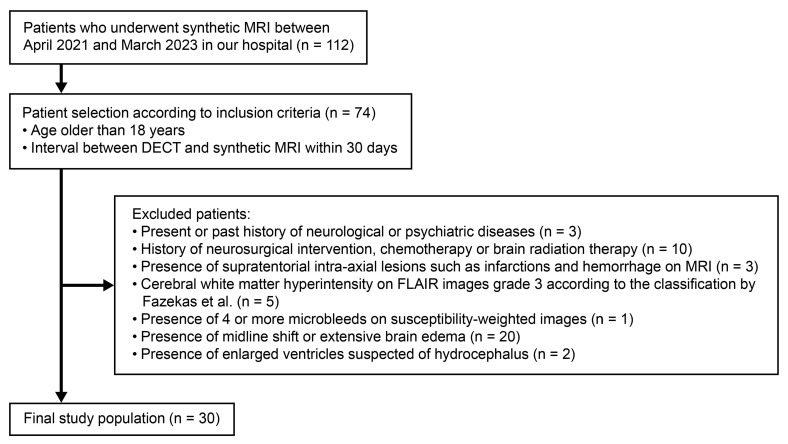
Flow of subject selection [23].

**Figure 3 tomography-11-00095-f003:**
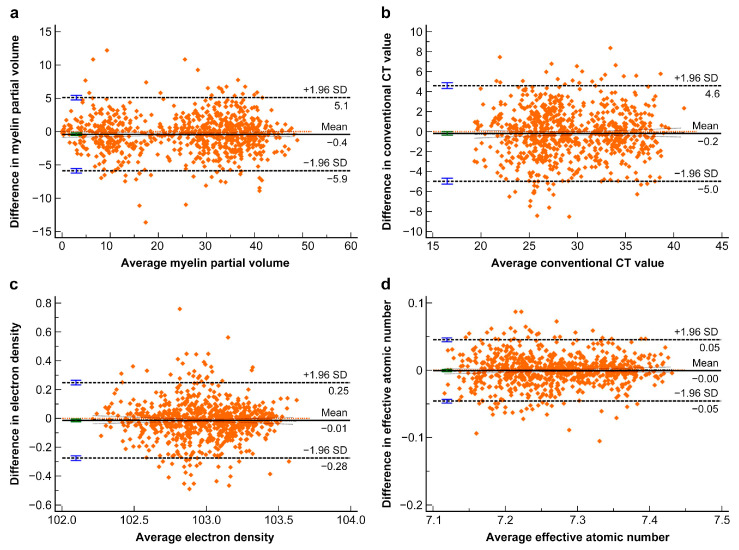
Bland–Altman plots for the myelin partial volume (**a**), conventional CT value (**b**), electron density (**c**), and effective atomic number (**d**) measurements.

**Figure 4 tomography-11-00095-f004:**
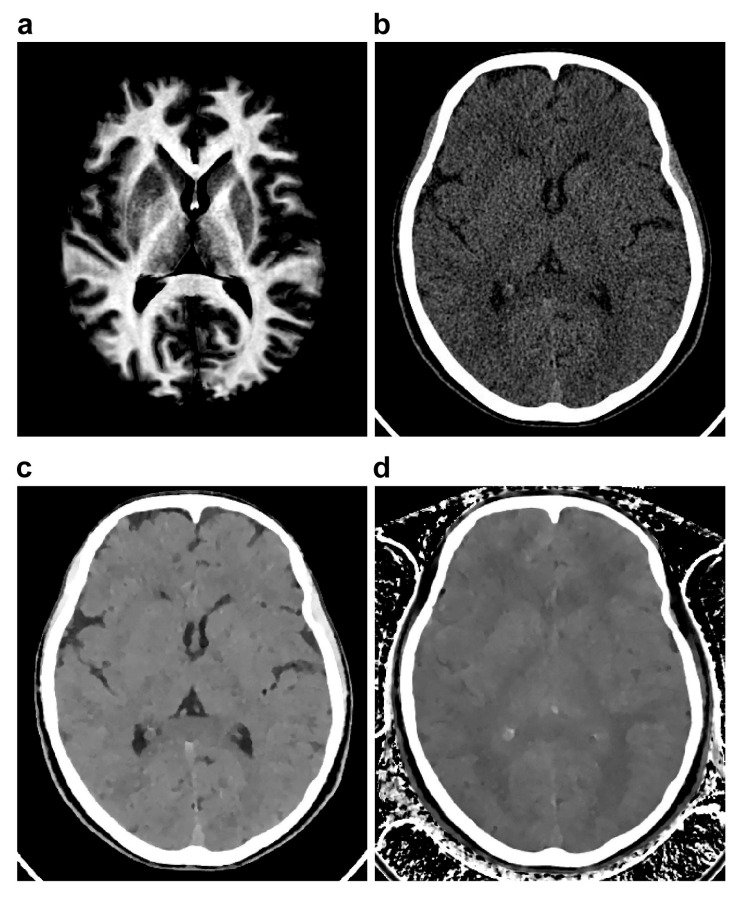
Representative images of the myelin partial volume (**a**), conventional CT (**b**), electron density (**c**), and effective atomic number (**d**).

**Figure 5 tomography-11-00095-f005:**
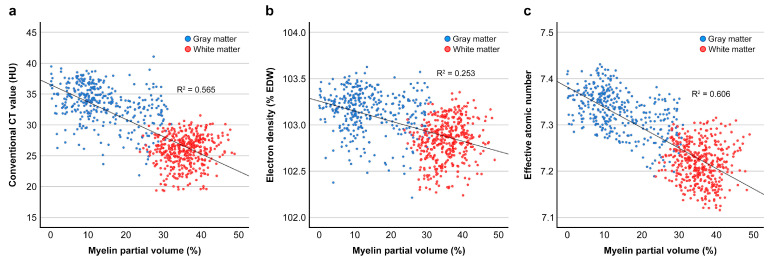
Correlation of the conventional CT value (**a**), electron density (**b**), and effective atomic number (**c**) with the myelin partial volume.

**Figure 6 tomography-11-00095-f006:**
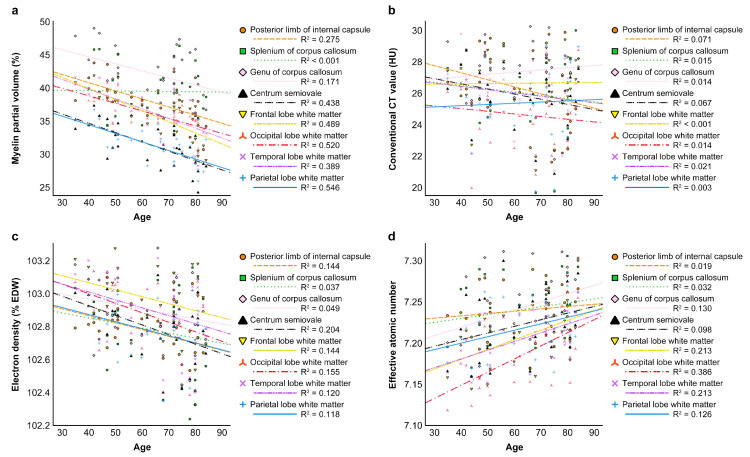
Age-related changes in the myelin partial volume (**a**), conventional CT value (**b**), electron density (**c**), and effective atomic number (**d**).

**Table 1 tomography-11-00095-t001:** Imaging parameters of conventional MRI sequences.

Parameter	FLAIR	SWI
Plane	Axial	Axial
TR (ms)	9000	31
TE (ms)	120	7.2/13.4/19.6/25.8
TI (ms)	2700	-
FA (°) (Refocus °)	180 (120)	17
Bandwidth (Hz/pixel)	322	255
Number of signal averages	1	4
Turbo factor	15	-
Acceleration factor	1.6	7
FOV (mm)	230 × 230	230 × 230
Matrix (frequency × phase)	511 × 511	767 × 767
Thickness (mm)	5	1
Slice number	24	150
Acquisition time (s)	162	116

FA, flip angle; FOV, field of view; TE, echo time; TR, repetition time.

**Table 2 tomography-11-00095-t002:** Locations of Regions of Interest (ROIs).

**White Matter (14 Regions)**	**Gray Matter (12 Regions)**
Right frontal lobe	Right caudate head
Left frontal lobe	Left caudate head
Right temporal lobe	Right putamen
Left temporal lobe	Left putamen
Right occipital lobe	Right globus pallidus
Left occipital lobe	Left globus pallidus
Right parietal lobe	Right medial thalamus
Left parietal lobe	Left medial thalamus
Right centrum semiovale	Right dorsal thalamus
Left centrum semiovale	Left dorsal thalamus
Right posterior limb of the internal capsule	Right lateral thalamus
Left posterior limb of the internal capsule	Left lateral thalamus
Genu of the corpus callosum	
Splenium of the corpus callosum	
**Cerebrospinal fluid (2 regions)**	
Right lateral ventricle	
Left lateral ventricle	

**Table 3 tomography-11-00095-t003:** Patient characteristics.

Characteristic	Value
Age	
Median	67.5 years
Range	35–84 years
30–39	1 (3%)
40–49	6 (20%)
50–59	4 (13%)
60–69	5 (17%)
70–79	8 (27%)
80–89	6 (20%)
Sex	
Female	22 (73%)
Indication of imaging studies	
Extra-axial tumor	23 (77%)
Infratentorial lesion	7 (23%)
Fazekas score	
Periventricular	
0	9 (30%)
1	16 (53%)
2	5 (17%)
Deep white matter	
0	17 (57%)
1	12 (40%)
2	1 (3%)
Number of microbleeds	
0	25 (83%)
1	3 (10%)
2	1 (3%)
3	1 (3%)

**Table 4 tomography-11-00095-t004:** Interobserver agreement.

	Intraclass Correlation Coefficients (95% CI)	Bland–Altman Analysis		
		Mean Differences (95% CI)	Lower Limit (95% CI)	Upper Limit (95% CI)
Myelin partial volume	0.975 (0.971, 0.979)	−0.398 (−0.595, −0.201)	−5.890 (−6.226, −5.553)	5.093 (4.756, 5.430)
Conventional CT value	0.873 (0.855, 0.889)	−0.174 (−0.3458, −0.000)	−4.963 (−5.256, −4.670)	4.615 (4.321, 4.908)
Electron density	0.884 (0.868, 0.899)	−0.013 (−0.023, −0.260)	−0.276 (−0.297, −0.260)	0.249 (0.233, 0.265)
Effective atomic number	0.948 (0.940, 0.955)	−0.000 (−0.002, 0.001)	−0.046 (−0.049, −0.043)	0.044 (0.043, 0.048)

95% CI, 95% confidence intervals.

**Table 5 tomography-11-00095-t005:** Myelin Partial Volume, Conventional CT Value, Electron Density, and Effective Atomic Number of White Matter, Gray Matter, and Cerebrospinal Fluid.

	WM (*n* = 420)	GM (*n* = 360)	CSF (*n* = 60)	*p* Value for Difference		
				WM vs. GM	GM vs. CSF	CSF vs. WM
Myelin partial volume (%)	35.6 ± 4.7	14.0 ± 8.1	0.0 ± 0.0	<0.001	<0.001	<0.001
Conventional CT value (HU)	25.9 ± 2.5	33.4 ± 3.2	4.1 ± 2.4	<0.001	<0.001	<0.001
Electron density (%EDW)	102.8 ± 0.2	103.2 ± 0.2	100.2 ± 0.2	<0.001	<0.001	<0.001
Effective atomic number	7.2 ± 0.0	7.3 ± 0.1	7.3 ± 0.1	<0.001	0.742	<0.001

CSF, cerebrospinal fluid; GM, gray matter; HU, Hounsfield unit; WM, white matter; %EDW, percent electron density relative to water. *p* values were obtained using the Mann–Whitney U test.

**Table 6 tomography-11-00095-t006:** Correlations between Myelin Partial Volume and CT parameters.

	All (WM + GM) (*n* = 780)		WM (*n* = 420)		GM (*n* = 360)	
	ρ	*p*	ρ	*p*	ρ	*p*
Conventional CT value	−0.705	<0.001	0.104	0.033	−0.379	<0.001
Electron density	−0.491	<0.001	0.202	<0.001	−0.151	0.004
Effective atomic number	−0.756	<0.001	−0.098	0.044	−0.478	<0.001

GM, gray matter; WM, white matter. Correlation was evaluated using Spearman’s rank correlation coefficient.

**Table 7 tomography-11-00095-t007:** Simple and multiple regression analyses of Myelin Partial Volume using CT parameters as independent variables.

		All (WM + GM) (*n* = 780)		WM (*n* = 420)		GM (*n* = 360)	
		R^2^	*p*	R^2^	*p*	R^2^	*p*
Simple regression analysis	Conventional CT value	0.565	<0.001	0.010	0.040	0.145	<0.001
	Electron density	0.253	<0.001	0.036	<0.001	0.027	0.002
	Effective atomic number	0.606	<0.001	0.005	0.133	0.252	<0.001
Multiple regression analysis	Electron density + Effective atomic number	0.675	<0.001	0.036	<0.001	0.290	<0.001

WM, white matter; GM, gray matter; R^2^, coefficient of determination.

**Table 8 tomography-11-00095-t008:** Correlations between patients’ age and Myelin Partial Volume or CT parameters.

	Myeline Partial Volume		Conventional CT Value		Electron Density		Effective Atomic Number	
	ρ	*p*	ρ	*p*	ρ	*p*	ρ	*p*
Caudate nucleus	−0.324	0.081	0.020	0.915	−0.119	0.532	0.184	0.331
Putamen	−0.276	0.139	−0.121	0.525	−0.205	0.276	0.283	0.130
Globus Pallidus	−0.646	<0.001	0.137	0.471	−0.166	0.380	0.275	0.142
Lateral thalamus	−0.605	<0.001	−0.001	0.995	−0.124	0.515	0.180	0.341
Medial thalamus	−0.378	0.040	0.047	0.807	−0.303	0.103	0.198	0.295
Dorsal thalamus	−0.334	0.071	0.308	0.098	0.034	0.858	0.211	0.264
Posterior limb of internal capsule	−0.552	0.002	−0.263	0.160	−0.413	0.023	0.149	0.433
Splenium of the corpus callosum	0.008	0.965	−0.009	0.964	−0.106	0.579	0.177	0.348
Genu of the corpus callosum	−0.413	0.023	0.165	0.383	−0.204	0.280	0.331	0.074
Centrum semiovale	−0.701	<0.001	−0.262	0.162	−0.450	0.013	0.264	0.158
Frontal lobe WM	−0.741	<0.001	0.016	0.931	−0.251	0.181	0.405	0.026
Occipital lobe WM	−0.678	<0.001	−0.148	0.436	−0.411	0.024	0.549	0.002
Temporal lobe WM	−0.649	<0.001	−0.199	0.293	−0.323	0.082	0.437	0.016
Parietal lobe WM	−0.719	<0.001	0.084	0.658	−0.305	0.101	0.308	0.098

WM, white matter. Correlation was evaluated using Spearman’s rank correlation coefficient.

## Data Availability

The original data presented in the study are openly available in FigShare at [https://doi.org/10.6084/m9.figshare.29595734].

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
