# Peer review of "Electron Density and Effective Atomic Number of Normal-Appearing Adult Brain Tissues: Age-Related Changes and Correlation with Myelin Content"

_tomography, 2025, doi:10.3390/tomography11090095_

Round 1
Reviewer 1 Report
Comments and Suggestions for Authors
The paper discusses the analysis of CT and MR images from 30 patients, from which it derives quantities such as Zeff, electron density, Hounsfield numbers, and myelin concentration.
The article is methodologically consistent. Due to its high number of numbers, it is difficult to read.
I suggest producing two additional tables.
First of all, since there are so many acronyms, a glossary that summarizes them all would be helpful.
It would also be helpful to include a table summarizing the points at which radiologists measure data with Image j.
It would also be helpful to discuss the dose released for acquisitions and whether the device undergoes regular quality control.
Finally, it would be appropriate to speculate on the clinical use of this approach.
Author Response
1. Summary |
|
|
Thank you very much for reviewing this manuscript. We have provided our responses to your comments in the space below. In addition, we have highlighted the corresponding revisions in the resubmitted Word file for your reference. We would greatly appreciate it if you could kindly review those changes as well. |
||
2. Point-by-point response to Comments and Suggestions for Authors |
|
|
Comments 1: there are so many acronyms, a glossary that summarizes them all would be helpful. |
Response 1: The list of abbreviations has been included in the manuscript (page 15, line 382 in the revised version). Would you prefer that I prepare a different version? |
|
Comments 2: It would also be helpful to include a table summarizing the points at which radiologists measure data with Image j. |
Response 2: Thank you very much for your valuable comment. We have added a summary table of the points where ROIs were placed. |
|
Comments 3: It would also be helpful to discuss the dose released for acquisitions and whether the device undergoes regular quality control. |
Response 3: Thank you for pointing this out. We agree with this comment. Therefore, we have added a statement in this study on page 3, line 105 of the revised manuscript. |
|
Comments 4: it would be appropriate to speculate on the clinical use of this approach. |
Response 4: I appreciate your insightful feedback. I agree with your observation. We have added a statement on page 14, line 321. |
Reviewer 2 Report
Comments and Suggestions for Authors
This paper presents data for electron density, effective atomic number, and relative myelin content appearing in normal adult brain tissue. It is performed as a retrospective study of 30 patients who underwent dual energy CT and synthetic MRI as part of their clinical workout. The data analysis focuses on investigating possible correlations between the various parameters, in particular when comparing gray matter and white matter. A significant negative correlation is obtained between myelin content and effective atomic number, while a somewhat weaker negative correlation is found between myelin content and electron density. Also the variation of the parameters with age is investigated. The work performed is generally scientifically sound.
The manuscript is in general well written, the methodological approach is well justified, and the results are clearly presented. The introduction includes a short overview of relevant previous work with references, however, the motivation for doing the current work seems rather weak and uses phrases like “the number of studies on ED and Zeff obtained from CT is limited” and “correlation with myelin partial volume and age-related changes have not been studied”. It is found that part of the introduction and discussion sections must be revised in order to present a stronger justification for doing the work, including the possible clinical relevance and impact of the results. Topics like radio therapy and the emerging photon counting CT technique may be considered.
The work is recommended for publication after minor revision.
Specific comments (page,line numbers):
P2, L74: Reference number for Institutional Review is incomplete
P3, L100-102: The Philips software used for calculation of ED and effective atomic number is crucial for this work, so please consider including a reference describing the methodology.
P3, L114-115: The software used for calculation of myelin partial volume is crucial for this work, so please consider including a reference describing the methodology.
P9, figure 5: Legends appear very small and difficult to read. The figure shows basically the same data as in table 4, so it may not add value to the manuscript.
P10, L238: Add “age” after “patient”.
P13, L287-289: In the possible explanation of the observed lower ED in WM, it may be relevant to include the mass density in the discussion. It is well established that adipose tissue has lower mass density than other soft tissues. See e.g. NIST Standard Reference Database 126 or ICRU report 44.
Author Response
1. Summary |
|
|
Thank you very much for reviewing this manuscript. We have provided our responses to your comments in the space below. In addition, we have highlighted the corresponding revisions in the resubmitted Word file for your reference. We would greatly appreciate it if you could kindly review those changes as well. |
||
2. Point-by-point response to Comments and Suggestions for Authors |
||
Comments 1: the motivation for doing the current work seems rather weak and uses phrases like “the number of studies on ED and Zeff obtained from CT is limited” and “correlation with myelin partial volume and age-related changes have not been studied”. It is found that part of the introduction and discussion sections must be revised in order to present a stronger justification for doing the work, including the possible clinical relevance and impact of the results.
|
||
Response 1: Thank you for pointing this out. We agree with this comment. Therefore, we have added the motivation for the study to the Introduction (page 2, line 53) and the possible clinical relevance to the Discussion section (page 14, line 321) in the revised manuscript attached.
|
||
Comments 2: P2, L74: Reference number for Institutional Review is incomplete |
||
Response 2: We sincerely apologize for the omission. We had not included the number, as we believed it was necessary to anonymize the manuscript for peer review. The number has now been added on page 2, line 78. |
||
Comments 3: P3, L100-102: The Philips software used for calculation of ED and effective atomic number is crucial for this work, so please consider including a reference describing the methodology. |
||
Response 3: We have added a reference related to the software methodology on page 3, line 112. |
||
Comments 4: P3, L114-115: The software used for calculation of myelin partial volume is crucial for this work, so please consider including a reference describing the methodology. |
||
Response 4: A reference regarding the software methodology has been added on page 3, line 127. |
||
Comments 5: P9, figure 5: Legends appear very small and difficult to read. The figure shows basically the same data as in table 4, so it may not add value to the manuscript. |
||
Response 5: Thank you for your comment. I agree with your point. Accordingly, Figure 5 from the original version of the manuscript has been removed. |
||
Comments 6: P10, L238: Add “age” after “patient”. |
||
Response 6: We sincerely apologize for the oversight. As you correctly noted, there was a missing word, which has now been corrected (page 11, line 248). |
||
Comments 7: P13, L287-289: In the possible explanation of the observed lower ED in WM, it may be relevant to include the mass density in the discussion. It is well established that adipose tissue has lower mass density than other soft tissues. See e.g. NIST Standard Reference Database 126 or ICRU report 44. |
||
Response 7: I appreciate your insightful feedback. Although ICRU Report 44 classifies brain tissue as a single category (Brain, Grey/White Matter) with uniform mass density (1.04 g/cm³), actual brain tissues exhibit regional compositional variations. White matter contains significantly higher lipid content (49-66% of dry weight) compared to gray matter (36-40%), primarily due to myelin-rich oligodendrocytes. Since lipids inherently possess lower mass density (~0.92 g/cm³) compared to proteinaceous components (~1.35 g/cm³), the higher lipid fraction in white matter contributes to both lower mass density and electron density relative to gray matter. This compositional difference may provide a fundamental physical basis for the observed ED variations between WM and GM. However, here we would like to limit ourselves to noting that the relative low ED of the WM is attributable to its abundant lipid content. |